# Mesh-Centric Gaussian Splatting for Human Avatar Modelling with Real-time Dynamic Mesh Reconstruction

Submission Id: 495

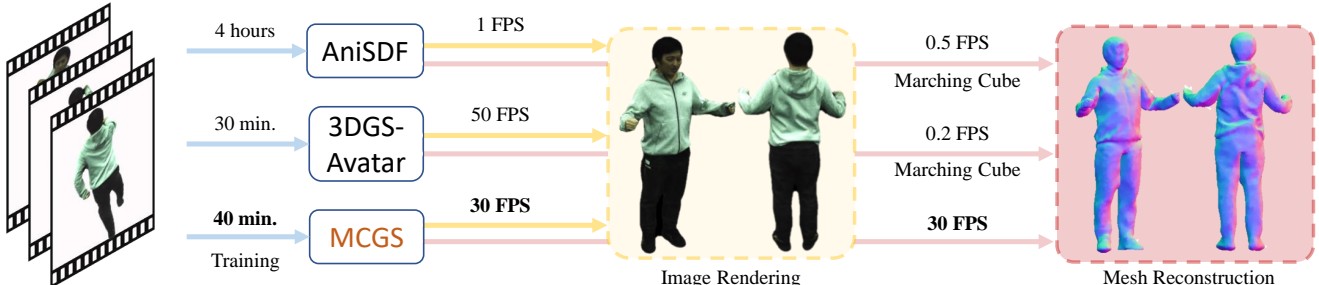

**Figure 1: Given a multi-view video, the proposed MCGS could be trained within 40 minutes and produce image rendering and mesh reconstruction both over 30 FPS without reliance on Marching Cube. In comparison, SDF-based method AniSDF [31] requires several hours of training with slow image rendering and mesh reconstruction. 3D Gaussian Splatting-based method 3DGS-Avatar [35] offers real-time image rendering but also falls behind on mesh reconstruction.**

## ABSTRACT

Real-time mesh reconstruction is highly demanded for integrating human avatar in modern computer graphics applications. Current methods typically use coordinate-based MLP to represent 3D scene as Signed Distance Field (SDF) and optimize it through volumetric rendering, relying on Marching Cubes for mesh extraction. However, volumetric rendering lacks training and rendering efficiency, and the dependence on Marching Cubes significantly impacts mesh extraction efficiency. This study introduces a novel approach, Mesh-Centric Gaussian Splatting (MCGS), which introduces a unique representation Mesh-Centric SDF and optimizes it using high-efficiency Gaussian Splatting. The primary innovation introduces Mesh-Centric SDF, a thin layer of SDF enveloping the underlying mesh, and could be efficiently derived from mesh. This derivation of SDF from mesh allows for mesh optimization through SDF, providing mesh as 0 iso-surface, and eliminating the need for slow Marching Cubes. The secondary innovation focuses on optimizing Mesh-Centric SDF with high-efficiency Gaussian Splatting. By dispersing the underlying mesh of Mesh-Centric SDF into multiple layers and generating Mesh-Constrained Gaussians on them, we create Multi-Layer Gaussians. These Mesh-Constrained Gaussians confine Gaussians within a 2D surface space defined by mesh, ensuring an accurate correspondence between Gaussian rendering and mesh geometry. The Multi-Layer Gaussians serve as sampling layers of Mesh-Centric SDF and can be optimized with Gaussian Splatting, which would further optimize Mesh-Centric SDF and its underlying mesh. As a result, our method can directly optimize the underlying mesh through Gaussian Splatting, providing fast training and rendering speeds derived from Gaussian Splatting, as well as precise surface learning of SDF. Experiments demonstrate that our method achieves dynamic mesh reconstruction at over 30 FPS. In contrast, SDF-based methods using Marching Cubes achieve less than 1 FPS, and concurrent 3D Gaussian Splatting-based methods cannot extract reasonable mesh.

## CCS CONCEPTS

• **Computing methodologies** → **Volumetric models**; *Animation*; *Mesh models*.

## KEYWORDS

3D Gaussian Splatting, neural implicit representation, volumetric rendering, novel view synthesis, dynamic motion, human shape and appearance modelling

**ACM Reference Format:**
Anonymous Author(s). 2024. Mesh-Centric Gaussian Splatting for Human Avatar Modelling with Real-time Dynamic Mesh Reconstruction. In *Proceedings of the 32nd ACM International Conference on Multimedia (MM '24), October 28 –November 1, 2024, Melbourne, Australia.* ACM, New York, NY, USA, 10 pages. https://doi.org/10.1145/nnnnnnn.nnnnnnn

## 1 INTRODUCTION

Human avatars play a pivotal role in a multitude of human-centric applications, ranging from movie production to telepresence and computer games. Modeling human figures, in contrast to static objects, presents distinct challenges due to their non-rigid dynamics, intricate poses, and diverse clothing variations. Intrigued by the potential applications of human avatars, numerous systems have been proposed to tackle these challenges [4, 9, 11–13]. However, many of these methods require the use of sophisticated equipment,

such as dense camera rigs [9, 12, 13], or controlled lighting setups [4, 11] for capturing human models. Subsequently, skilled artists are often tasked with manually designing a skeleton for the human model and meticulously adjusting skinning weights to achieve realistic animations [16]. Consequently, the substantial investment in equipment and human labor confines the applicability of these methods to research laboratories or larger corporations.

Recent advancements in neural implicit representations have significantly reduced cost of creating human avatars by leveraging multi-view videos [8, 17, 30, 30, 45, 46, 52]. These methods typically involve learning a deformation field [8, 17, 30, 30, 45] to deform a static implicit model [25, 43] or directly modeling a dynamic implicit model [46, 52]. Some methods [31, 44] further integrate Signed Distance Fields (SDF) as implicit model to leverage its surface constraint, demonstrating promising mesh reconstruction performance. Despite their capabilities, these approaches lag in efficiency, primarily due to the inherent limitations of volumetric rendering in implicit models. Volumetric rendering requires querying dense sample points along a camera ray to simulate density distribution, leading to a high number of queries for the entire scene [23]. To address this efficiency issue, some methods have incorporated specialized data structures like Plenoctrees [49], K-Planes [6], or Hash Encodings [38] into human avatar creation [8]. However, these methods are still tethered to volumetric rendering and suffer from excessive querying.

In parallel with the evolution of implicit representations, explicit representations remain prevalent in the industry, encompassing Mesh [15], Voxels [20, 40], Point Clouds [1, 5], and Multiplane Images (MPI) [24, 53]. Notably, a Point Cloud-based technique called 3D Gaussian Splatting (3DGS) [14] has gained popularity for capturing 3D scenes and rendering them from various viewpoints. This method represents a scene using numerous small 3D Gaussians with optimized positions, orientations, appearances (represented as spherical harmonics). Rendering involves splatting a limited number of optimized Gaussians onto the image plane, offering a faster alternative to implicit representations. While 3DGS enables realistic renderings, leveraging this technique for human avatar creation poses challenges, particularly in extracting mesh from the Gaussians due to their unordered structure and lack of correspondence with the scene's actual surface. Failure in mesh extraction hinders its utilization in numerous applications, encompassing texture editing, sculpting, animation, and Physically Based Rendering.

In this paper, we present a method achieving real-time mesh reconstruction for human avatars by optimizing Mesh-Centric Signed Distance Field (SDF) with high-efficiency Gaussian Splatting. The first innovation introduces Mesh-Centric SDF, which is a thin layer of SDF enveloping the underlying mesh, where Signed Distance for a specific point is its distance from the underlying mesh. Mesh-Centric SDF could be efficiently derived from mesh, allowing for mesh optimization through SDF optimization. It naturally provides mesh as 0 iso-surface, eliminating the need for slow Marching Cubes. The secondary innovation focuses on optimizing Mesh-Centric SDF with high-efficiency Gaussian Splatting. By dispersing the underlying mesh of Mesh-Centric SDF into multiple layers and generating Mesh-Constrained Gaussians on them, we create Multi-Layer Gaussians. Mesh-Constrained Gaussians involves attaching a set of flat Gaussians to triangle faces of mesh, with rotations and

scaling constrained within the triangle plane. This flat representation ensures an even distribution over mesh surface, effectively aligning rendering of Gaussians with geometry of mesh. The Multi-Layer Gaussians serve as sampling layers of Mesh-Centric SDF and can be optimized with Gaussian Splatting, which would further optimize Mesh-Centric SDF and its underlying mesh. As a result, the proposed method, Mesh-Centric Gaussian Splatting (MCGS), can directly optimize the underlying mesh through efficient Gaussian Splatting, combining fast training and rendering speed of Gaussian Splatting with precise surface learning of SDF. Experimental results demonstrate that our method achieves dynamic mesh retrieval at over 30 FPS. In comparison, SDF-based methods using Marching Cubes achieve less than 1 FPS, and concurrent 3D Gaussian Splatting methods cannot extract reasonable mesh. The real-time mesh reconstruction capability will further enable seamless integration with established computer graphics pipelines for editing, sculpting, animation, and Physically Based Rendering, thereby enhancing industry engagement.

In summary, our contributions include:

- Proposing Mesh-Centric Gaussian Splatting (MCGS) for human avatar modelling with real-time mesh reconstruction, combining fast training and rendering of Gaussian Splatting with precise surface learning of SDF.
- Introducing Mesh-Centric SDF as a novel surface representation, which is a thin layer of SDF wrapped around mesh and naturally provides mesh as 0 iso-surface, eliminating reliance on slow Marching Cubes to extract 0 iso-surface.
- Optimizing Mesh-Centric SDF with high-efficiency Gaussian Splatting, achieved by generating Multi-Layer Gaussians from underlying mesh of Mesh-Centric SDF and optimizing them with Gaussian Splatting.

## 2 RELATED WORKS

### 2.1 Implicit Human Avatar Modelling Methods

Starting from Neural Radiance Fields (NeRF) [23], implicit representations have sparked a revolution in traditional reconstruction methods [7, 18, 23, 25, 43, 47, 48, 50]. These methods depict a scene as continuous functions, offering advantages in memory efficiency and high resolution. For instance, NeRF [25] directly maps a continuous 5D coordinate (including spatial coordinate and viewing direction) to volume density and view-dependent emitted radiance, serving as a successful example of representing a scene as a neural radiance field. NeRF can portray a continuous scene in arbitrary resolution and effectively learn from multi-view images using a differentiable volumetric renderer.

Recent studies have integrated implicit representation into dynamic human modeling by combining a static neural implicit representation with dynamic deformation fields [19, 27, 28, 30, 32, 34, 39, 45]. These methods typically utilize SMPL [21] as a prior for skeleton-based motion, with deformation fields mainly representing non-linear deformations. Another approach involves directly learning dynamic implicit representation that conditions the neural implicit representations on human pose [46, 52]. These methods demonstrate a robust representation of pose-dependent dynamics. Some methods [31, 44] further integrate Signed Distance Fields

(SDF) as implicit model to leverage its surface constraint, demonstrating promising mesh reconstruction performance.

However, implicit representations have certain limitations. Firstly, they rely on volumetric rendering, which defines neural fields throughout the entire 3D space, even if the 3D object only occupies a small fraction of that space, leading to memory redundancy. Secondly, volumetric rendering requires dense sample points along a ray to accumulate as pixel color, resulting in significant computation. In contrast, explicit representations represent scenes as explicit components, providing the potential to directly project components into images with real-time efficiency. The proposed method in this work is based on an explicit representation called 3D Gaussian Splatting, which enhances real-time rendering efficiency.

## 2.2 Explicit Human Avatar Modelling Methods

Explicit representations, such as mesh [15], voxels [20, 40], point clouds [1, 5], and multiplane images (MPI) [24, 53], continue to be the mainstream in production due to their high efficiency and ease of manipulation, allowing for immediate interactions like texture editing. However, developing explicit models is often costly, involving manual design that requires significant labor. To mitigate these costs, parametric models [2, 21, 29] are commonly used to fit parametric body models directly to skinned human scans. Nevertheless, these models are not suitable for modeling clothed humans, which are much more complex and require expensive 3D scans.

Recently, 3D Gaussian Splatting (3DGS) [14] has emerged as an efficient alternative to implicit representation for fast inference and training with point-based rendering. This approach models the rendering process as splatting a set of 3D Gaussians onto the image plane via alpha blending, achieving state-of-the-art rendering quality with real-time inference speed and rapid training when provided with multi-view inputs. However, 3DGS is not currently compatible with existing computer graphics pipelines, and the process of extracting a mesh from millions of tiny Gaussians after optimization remains unclear.

In this study, we introduce a method that achieves high-fidelity rendering and real-time mesh reconstruction using 3D Gaussian Splatting (3DGS). This innovative approach paves the way for integrating 3DGS into modern computer graphics pipelines. While some concurrent works [10, 51] also explore extracting meshes from 3DGS, they exhibit certain gaps compared to our method. These works typically involve jointly learning an implicit Signed Distance Field (SDF) and explicit Gaussians, and utilize isolated methods such as Marching Cube or Poisson Reconstruction to extract meshes from SDF or point clouds. However, this strategy relies on Marching Cube or Poisson Reconstruction, and the process of mesh extraction consistently operates at a speed slower than 1 FPS. In contrast, our method directly optimizes the underlying mesh by optimizing Gaussians, achieving high reconstruction accuracy and demonstrating promising real-time efficiency.

## 3 PROPOSED METHOD

In line with previous works targeting surface reconstruction for human avatars [31, 44], we utilize a training set of $T$-frame multi-view videos of a human performer captured by a sparse set of $K$ synchronized and calibrated cameras: $I = \{I_t^k\}$ ($t = 1 \ldots T, k = 1 \ldots K$). We aim at human avatar modelling with real-time mesh reconstruction and the ability to perform actions not present in training frames $I$. We employ off-the-shelf methods to derive segmentation masks and fit parametric body models (SMPL [21] in this work).

The pipeline of proposed method is illustrated in Figure 2. We begin with T-pose Canonical Mesh, which is initialized as SMPL mesh and deformed using non-linear deformation $f_{non-linear}$ and Linear Blending Skinning (LBS) to generate target-pose Deformed Mesh, as explained in Section 3.2. The Deformed Mesh serves as dynamic mesh reconstruction based on pose estimation of SMPL. Subsequently, we construct Mesh-Centric SDF around Deformed Mesh and disperse Deformed Mesh into Multi-Layer Mesh along its vertex normal vectors, serving as sampling layers of Mesh-Centric SDF, as depicted in Section 3.3. The Mesh-Centric SDF, a volume representation, offers spatial coverage of Real Surface and facilitates surface learning similar to previous SDF-based methods. To efficiently train it using Gaussian Splatting, we store Primitives $P$ on triangle faces of T-pose Canonical Mesh and deform them with mesh deformation. These Primitives remain fixed during mesh deformation and are subsequently transformed into Multi-Layer Gaussians, as elaborated in Section 3.4. Finally, we employ Gaussian Splatting described in Section 3.1 to optimize Multi-Layer Gaussians, which further optimizes non-linear deformation $f_{non-linear}$, LBS, T-pose Canonical Mesh, and Primitives attached to it.

## 3.1 Prelininary: 3D Gaussian Splatting

3D Gaussian Splatting (3DGS) [14] captures a 3D scene using an ensemble of 3D Gaussians $\{G\}$, each characterized by position $\mathbf{x}$, covariance $\Sigma$, opacity $\alpha$, and color $\mathbf{c}$ represented by spherical harmonics (SH). To ensure positive semi-definiteness, the covariance matrix $\Sigma$ is represented by scaling matrix $\mathbf{S}$ and rotation matrix $\mathbf{R}$:

$$\Sigma = \mathbf{R}\mathbf{S}\mathbf{S}^T\mathbf{R}^T. \tag{1}$$

Practically, 3DGS stores diagonal vector $\mathbf{s} \in \mathbb{R}^3$ of the scaling matrix and a quaternion vector $\mathbf{q} \in \mathbb{R}^4$ to represent rotation matrix, which can be converted to a valid covariance matrix.

To optimize from image observation, 3DGS builds a differentiable Gaussian Splatting process to render 3D Gaussians into image. Specifically, 3D Gaussians are projected onto image plane, and alpha blending is employed to compute pixel color $C$ by blending $N$ ordered Gaussians that overlap the pixel:

$$C = \sum_{i \in N} \mathbf{c}_i \alpha_i' \prod_{j=1}^{i-1} (1 - \alpha_j'), \tag{2}$$

where $\alpha_i'$ denotes the learned opacity $\alpha_i$ weighted by the probability density of the $i$-th projected Gaussian at the target pixel location.

## 3.2 Mesh Deformation

We build an efficient method to obtain pose-dependent mesh reconstruction given pose estimation of SMPL. Illustrated in Figure 2, the T-pose Canonical Mesh is initialized as SMPL mesh. For vertex $\mathbf{v}_c$ on Canonical Mesh, we deform it with pose-dependent non-linear deformation to obtain $\mathbf{v}_n$. The non-linear deformation is estimated by a hash-encoding MLP [38] $f_{non-linear}$:

$$\mathbf{v}_n = \mathbf{v}_v + f_{non-linear}(\mathbf{v}_c, \boldsymbol{\theta}), \tag{3}$$

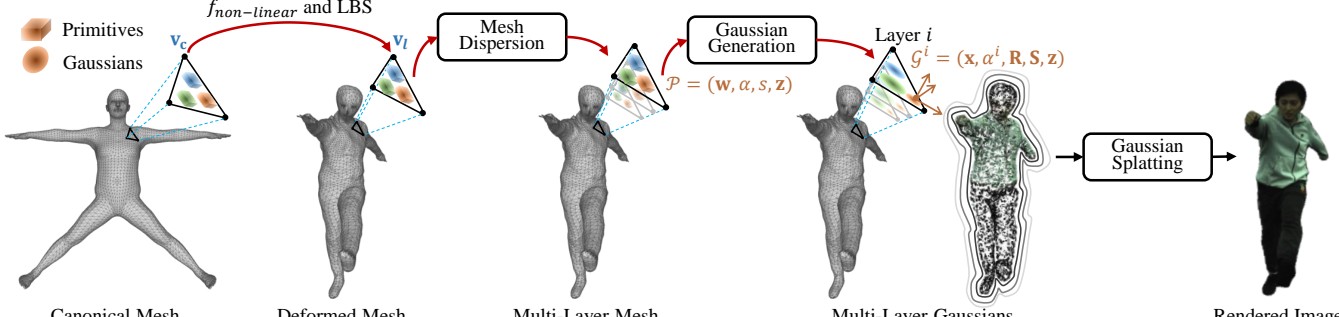

**Figure 2: Pipeline of proposed MCGS. We deform T-pose Canonical Mesh with non-linear and linear deformations to obtain Deformed Mesh at target pose (Section 3.2). Subsequently, we disperse Deformed Mesh to Multi-Layer Mesh, acting as sampling layers of Mesh-Centric SDF (Section 3.3). To optimize Mesh-Centric SDF, we store Primitives $\mathcal{P}$ on triangle faces of Canonical Mesh and deform them with mesh deformation. Primitives are further transformed into Multi-Layer Gaussians using Gaussian Generation described in Section 3.4. Finally, Gaussian Splatting is utilized to optimize Multi-Layer Gaussians with image observation, which further optimizes Primitives, Canonical Mesh, non-linear deformation, and linear deformation.**

where $\boldsymbol{\theta}$ is pose parameters of SMPL. Then $\mathbf{v}_n$ is transformed to target-pose Deformed Mesh using Linear Blending Skinning (LBS):

$$\mathbf{v}_l = \text{LBS}(f_{skinning}(\mathbf{v}_n); \boldsymbol{\theta}, \boldsymbol{\beta}), \qquad (4)$$

where $\boldsymbol{\theta}$ is pose parameters of SMPL, $\boldsymbol{\beta}$ is shape parameters of SMPL, and a skinning MLP $f_{skinning}$ is learned to predict skinning weight of $\mathbf{v}_n$. During inference, we could efficiently obtain pose-dependent mesh reconstruction by deforming vertices with non-linear deformation $f_{non-linear}$ and LBS.

As we possess target-pose Deformed Mesh as mesh reconstruction result, we could conduct mesh smoothness loss on Deformed Mesh to regularize mesh deformation. Specifically, we incorporate regularization losses from Pytorch3d [36] concerning edge length, normal consistency, and Laplacian smoothing. Our full loss function consists of a RGB loss $L_{RGB}$, a mask loss $L_{mask}$, a skinning weight regularization loss $L_{skin}$ and a mesh smoothness loss $L_{mesh}$. More details are shown in Supplementary Material.

## 3.3 Mesh Dispersion

In this section, we describe the definition of Mesh-Centric SDF and how to generate Multi-Layer Mesh using Mesh Dispersion. As depicted in Figure 3 (a), we take Deformed Mesh from Section 3.2 as underlying mesh and build Mesh-Centric Signed Distance Field (SDF) as a thin layer of SDF enveloping the underlying mesh, with the mesh itself serving as 0 iso-surface. Naturally, the Signed Distance of a point corresponds to its distance from the underlying mesh. Since Mesh-Centric SDF is derived from the underlying mesh, we can directly optimize the underlying mesh through SDF optimization, eliminating the need for time-consuming Marching Cube algorithm [22] typically used to extract 0 iso-surface in previous SDF-based methods [31, 43, 44].

Traditional SDF-based methods typically represent SDF as a Coordinate-based MLP and utilize volumetric rendering for optimization, which involves extensive querying of the MLP and significantly hinders efficiency. The proposed Mesh-Centric SDF naturally incorporates the underlying mesh as 0 iso-surface, allowing for optimization through a more efficient Mesh Dispersion strategy. Details of Mesh Dispersion are illustrated in Figure 3 (a).

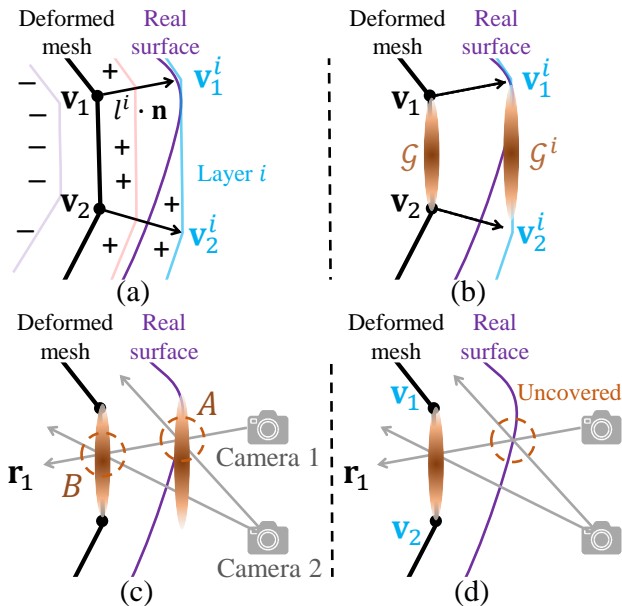

**Figure 3: 2D example of: (a) Mesh-Centric SDF and Mesh Dispersion in Section 3.3, (b) Multi-Layer Gaussians generated in Section 3.4, (c) Optimization process of Mesh-Centric SDF, (d) Optimization process of naive Mesh-Constrained Gaussians and previous works [10, 42].**

By dispersing the underlying mesh into multiple layers along its vertex normal vectors $\mathbf{n}$ at random distances, we obtain Multi-Layer Mesh acting as sampling layers of SDF enveloping the underlying mesh, among which layer $i$ with Signed Distance $l^i$ approximately covers Real Surface. In the next section, we would describe how to generate Gaussians on Multi-Layer Mesh and reweight them with respective Signed Distance, forming Multi-Layer Gaussians in Figure 3 (b). Thus we could optimize Signed Distance values of Multi-Layer Mesh through Gaussian Splatting and further optimize the underlying mesh.

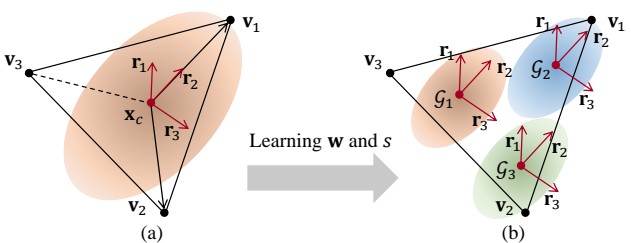

**Figure 4: Generating Mesh-Constrained Gaussians from Primitives. (a) We calculate rotation matrix $\mathbf{R} = (\mathbf{r}_1, \mathbf{r}_2, \mathbf{r}_3)$ and scaling matrix S from vertices $(\mathbf{v}_1, \mathbf{v}_2, \mathbf{v}_3)$. (b) With learnable w and $s$, several Gaussians are set on a same triangle to represent more details.**

### 3.4 Gaussian Generation

Naive Gaussian Splatting in Section 3.1 achieves high efficiency and high-quality rendering. However, it exhibits poor alignment between appearance and geometry, as highlighted in previous studies [10, 42]. To mitigate this issue, we confine Gaussians to a 2D surface space defined by an underlying mesh, ensuring a consistent correspondence between appearance and geometry. This methodology enables us to achieve precise surface reconstruction by optimizing appearance through alpha blending in equation (2). In Figure 2, we store Primitives $\mathcal{P}$ on triangle faces of mesh and convert them into Gaussians $\mathcal{G}$. We start with a basic scenario where we generate Gaussian $\mathcal{G}$ from Primitive $\mathcal{P}$ attached to triangle face $V$. Subsequently, we elaborate on the process of generating Multi-Layer Gaussians, as depicted in Figure 3 (b).

We then describe converting Primitive $\mathcal{P} = (\mathbf{w}, \alpha, s, \mathbf{z})$ on triangle face $V = (\mathbf{v}_1, \mathbf{v}_2, \mathbf{v}_3)$ to Mesh-Constrained Gaussian $\mathcal{G} = (\mathbf{x}, \alpha, \mathbf{R}, \mathbf{S}, \mathbf{z})$. For each Primitive, $\mathbf{w} = (w_1, w_2, w_3)$ is barycentric coordinates and $w_1 + w_2 + w_3 = 1$, $\alpha$ is opacity, $s$ is scale relative to triangle face, and $\mathbf{z}$ is multi-dimension color feature. To capture appearance dynamics like shadows, we follow a similar approach to 3DGS-Avatar [35] by storing per-Primitive color feature vectors $\mathbf{z}$ and employing a small pose-dependent Color MLP $f_{color}$ to predict Gaussian colors $\mathbf{c}$: $\mathbf{c} = f_{color}(\mathbf{z}, \boldsymbol{\theta})$, where $\boldsymbol{\theta}$ denotes pose parameters of SMPL. Given the underlying mesh, we possess knowledge of vertex order and define rotation and scaling matrices relative to triangle face. We define rotation matrix $\mathbf{R}$ in equation 1 as orthonormal vectors: $\mathbf{R} = [\mathbf{r}_1, \mathbf{r}_2, \mathbf{r}_3]$. As shown in Figure 4, the first vector $\mathbf{r}_1$ is defined as the normal vector:

$$\mathbf{r}_1 = \frac{(\mathbf{v}_2 - \mathbf{v}_1) \times (\mathbf{v}_3 - \mathbf{v}_1)}{||(\mathbf{v}_2 - \mathbf{v}_1) \times (\mathbf{v}_3 - \mathbf{v}_1)||}. \tag{5}$$

The second vector $\mathbf{r}_2$ is defined as the vector from triangle center $\mathbf{x}_c$ to vertex $\mathbf{v}_1$:

$$\mathbf{r}_2 = \frac{\mathbf{v}_1 - \mathbf{x}_c}{||\mathbf{v}_1 - \mathbf{x}_c||}, \tag{6}$$

where $\mathbf{x}_c = \text{mean}(\mathbf{v}_1 + \mathbf{v}_2 + \mathbf{v}_3)$ is triangle center. Finally, the third vector $\mathbf{r}_3$ is obtained through orthonormalizing the vector with respect to the existing two vectors (a single step in the Gram–Schmidt process [3]):

$$\mathbf{r}_3 = \frac{\text{orth}(\mathbf{v}_2 - \mathbf{x}_c; \mathbf{r}_1, \mathbf{r}_2)}{||\text{orth}(\mathbf{v}_2 - \mathbf{x}_c; \mathbf{r}_1, \mathbf{r}_2)||}, \tag{7}$$

where

$$\text{orth}(\mathbf{x}; \mathbf{r}_1, \mathbf{r}_2) = \mathbf{x} - \text{proj}(\mathbf{x}; \mathbf{r}_1) - \text{proj}(\mathbf{x}; \mathbf{r}_2), \tag{8}$$

and

$$\text{proj}(\mathbf{v}, \mathbf{u}) = \frac{\langle \mathbf{v}, \mathbf{u} \rangle}{\langle \mathbf{u}, \mathbf{u} \rangle} \mathbf{u}. \tag{9}$$

For scaling parameters **S**, we use:

$$\mathbf{S} = \text{diag}(s_1, s_2, s_3), \tag{10}$$

where $s_1 = \epsilon$, $s_2 = ||\mathbf{x}_c - \mathbf{v}_1||$, and $s_3 = \langle \mathbf{v}_2, \mathbf{r}_3 \rangle$. $s_1$ corresponds with the normal vector and is fixed to be a constant small value $\epsilon$.

Consequently, the covariance $\Sigma$ of mesh-constrained Gaussian is obtained through equation (1) and corresponds with the shape of triangle $V$. To represent more details, as shown in Figure 4 (b), we store several Primitives $\mathcal{P} = (\mathbf{w}, \alpha, s, \mathbf{z})$ on a same triangle face and control their position with learnable barycentric coordinate $\mathbf{w}$ by:

$$\mathbf{x} = w_1 \mathbf{v}_1 + w_2 \mathbf{v}_2 + w_3 \mathbf{v}_3, \tag{11}$$

and control their size with a learnable scale parameters $s$:

$$\mathbf{S}' = \text{sigmoid}(s)\mathbf{S}. \tag{12}$$

Subsequently, we describe the process of generating Multi-Layer Gaussians on Multi-Layer Mesh, where their opacities are adjusted based on respective Signed Distances. As illustrated in Figure 3 (b), the opacity for $\mathcal{G}^i$ on layer $i$ is initially inherited from Primitive $\mathcal{P}$ and then further modified by Signed Distance $l^i$ between layer $i$ and the underlying mesh. To be specific, we convert Signed Distance $l^i$ into density $\sigma^i$ in accordance with [43]:

$$\sigma^i = \begin{cases} \dfrac{1}{\beta}(1 - \dfrac{1}{2}\exp(\dfrac{l^i}{\beta})), & \text{if } l^i < 0, \\[2ex] \dfrac{1}{2\beta}\exp(-\dfrac{l^i}{\beta}), & \text{if } l^i \geq 0, \end{cases} \tag{13}$$

where $\beta$ is an optimizable parameter controlling the degree of density concentration when converting SDF to density field. Then we adjust opacity $\alpha$ with $\sigma^i$:

$$\alpha^i = \sigma^i \cdot \alpha. \tag{14}$$

***Remark.*** There have been several methods that involve attaching flat Gaussians onto meshes [10, 42]; however, these methods rely on pre-captured meshes and struggle to learn accurate geometry. The proposed Mesh-Centric SDF is pivotal in learning precise geometry and distinguishes our approach from these existing techniques. In Figure 3 (c), a 2D example illustrates the learning process of Mesh-Centric SDF. Multiple positions are optimized by ray $\mathbf{r}_1$ from camera 1, encompassing point $A$ near Real Surface and point $B$ farther away. When optimized with rays from camera 2, point $A$ near Real Surface is assigned a higher blending weight, which enhances $\alpha^i$ and $\sigma^i$ in equation (14). Consequently, the Signed Distance $l^i$ is reduced, bringing the underlying mesh closer to Real Surface. In contrast, the naive Mesh-Constrained Gaussians and previous methods that involve attaching flat Gaussians onto meshes [10, 42] are depicted in Figure 3 (d). These methods only consider a single layer of Gaussians and optimize a single position from ray $\mathbf{r}_1$, resulting in a lack of spatial coverage of Real Surface. Consequently, the optimization process falls short in accurately learning geometry by failing to enhance the alpha blending weight at the intersection point of ray $\mathbf{r}_1$ and Real Surface.

**Table 1: Efficiency Comparison including training time (GPU hours), image rendering speed (FPS), and mesh reconstruction speed (FPS).**

|  | Neural Body | AniSDF | 3DGS-Avatar | **Ours** |
|---|---|---|---|---|
| Training(GPU) | 5h | 7h | 0.5h | **0.7h** |
| Rendering(FPS) | 3.5 | 0.9 | 50 | **32** |
| Reconstruction(FPS) | 1.2 | 0.4 | 0.2 | **32** |

## 4 EXPERIMENT

### 4.1 Datasets and Metrics

**ZJU-MoCap** [33] records multi-view videos with 21 synchronous cameras and collects shape parameters of SMPL as well as global translation and SMPL's pose parameters with an off-the-shelf SMPL tracking system [54]. Following [31], we choose 4 uniformly distributed cameras for training and the remaining cameras for testing. On ZJU-MoCap dataset, we evaluate image rendering performance on seen poses and unseen poses with PSNR and LPIPS metrics.

**SyntheticHuman** [31] is a synthetic dataset that contains 7 animated 3D characters from RenderPeople [37] and Mixamo [26], which provides 3D ground truth. Similarly, we choose 4 cameras for training and the remaining cameras for testing. We use this dataset to evaluate the performance of 3D reconstruction and adopt Chamfer Distance (CD) and Point-To-Surface Euclidean distance (P2S) as metrics. Units for CD and P2S are in $cm$.

### 4.2 Compared Methods

The compared methods include NeuralBody [33], AniSDF [31], and concurrent 3DGS-Avatar [35], which are SOTA methods based on Neural Radiance Field (NeRF), Signed Distance Fields (SDF), and 3D Gaussian Splatting (3DGS), respectively. It is worth noting that 3DGS-Avatar, which is based on 3DGS, struggles to generate reasonable meshes. Therefore, our comparative evaluation with 3DGS-Avatar is focused on image rendering performance, as elaborated in Section 4.5. For completeness, we include the mesh reconstruction results of 3DGS-Avatar in the Supplementary Material.

### 4.3 Performance on Efficiency

The primary strength of our method lies in its exceptional efficiency in mesh reconstruction. During inference, we leverage $f_{\text{non-linear}}$ and LBS in equations (3) and (4) to rapidly generate pose-dependent meshes from Canonical Mesh, enabling real-time mesh reconstruction. Table 1 clearly demonstrates that compared methods significantly lag in mesh rendering efficiency. This performance gap arises from their approach of voxelizing the density field and using Marching Cubes [22] for mesh extraction. For 3DGS-Avatar, we apply mesh extraction methods proposed in DreamGaussian [41]. However, we observed that it fails to generate reasonable meshes when applied to human avatars, leading us to present its mesh reconstruction results in the Supplementary Material.

Certain concurrent works [10, 51] concentrate on training SDF and Gaussian Splatting together for static scenes, employing voxelization and Marching Cubes for mesh extraction, which encounter similar speed limitations and have yet to explore in human avatar modelling scenarios. In contrast, our method directly optimizes

**Table 2: Quantitative results of mesh reconstruction on SyntheticHuman dataset.**

|  | P2S ↓ | | | CD ↓ | | |
|---|---|---|---|---|---|---|
|  | Neural Body | AniSDF | OURS | Neural Body | AniSDF | OURS |
| S1 | 1.29 | 0.62 | **0.49** | 1.21 | 0.65 | **0.52** |
| S1 | 1.20 | 0.67 | **0.36** | 1.12 | 0.66 | **0.37** |
| S3 | 1.60 | 0.93 | **0.40** | 1.67 | 1.24 | **0.67** |
| S4 | 0.98 | 0.59 | **0.27** | 1.11 | 0.74 | **0.44** |
| S5 | 0.98 | 0.42 | **0.32** | 0.99 | 0.54 | **0.41** |
| S6 | 0.87 | 0.54 | **0.28** | 1.02 | 0.78 | **0.51** |
| S7 | 0.80 | 0.35 | **0.26** | 0.99 | 0.50 | **0.40** |
| Avg. | 1.10 | 0.59 | **0.34** | 1.16 | 0.73 | **0.47** |

the underlying mesh through Gaussian Splatting, eliminating reliance on Marching Cubes and ensuring real-time efficiency in mesh reconstruction.

Another notable strength of our approach is its efficiency in both training and image rendering, a merit derived from leveraging 3D Gaussian Splatting. While 3DGS-Avatar also leverages 3D Gaussian Splatting and demonstrates efficiency in training and rendering, it grapples with the challenge of mesh extraction from 3D Gaussians, hindering its mesh generation capabilities.

### 4.4 Performance on Mesh Reconstruction

In addition to significantly enhanced mesh reconstruction efficiency, quantitative results presented in Table 2 demonstrate that our method outperforms compared approaches by a substantial margin in terms of accuracy. Through a qualitative assessment depicted in Figure 5, we can observe distinct characteristics of compared methods. NeuralBody generates noisy meshes due to the absence of surface constraints, whereas AniSDF yields smoother meshes by leveraging surface constraint of SDF. However, AniSDF tends to introduce noise in certain body parts, as evidenced by the arm of the left subject in the second row. This issue can be attributed to the utilization of SDF as an intermediary for surface representation and further adopts Marching Cube to extract mesh. In contrast, our approach learns mesh directly without relying on Marching Cube, resulting in a higher consistency with Ground Truth.

### 4.5 Performance on Image Rendering

Benefiting from representation capabilities of 3D Gaussian Splatting, our method demonstrates outstanding image rendering quality. We assess image rendering performance on both seen and unseen poses using ZJU-MoCap [33] dataset. As highlighted by Nerfies [28], PSNR metric tends to favor blurry images; therefore, we emphasize the LPIPS metric in our evaluation. Quantitative analysis in Table 3 demonstrates that 3DGS-based methods, including 3DGS-Avatar and our method, achieve significantly better LPIPS metrics compared to Neural Body and AniSDF, showcasing the enhanced representation abilities of 3D Gaussian Splatting. Furthermore, our method exhibits superior performance compared to 3DGS-Avatar due to the additional surface constraints introduced by constraining Gaussians on the mesh. In terms of the PSNR metric, the differences are relatively minor, with our proposed method performing competitively with the compared approaches.

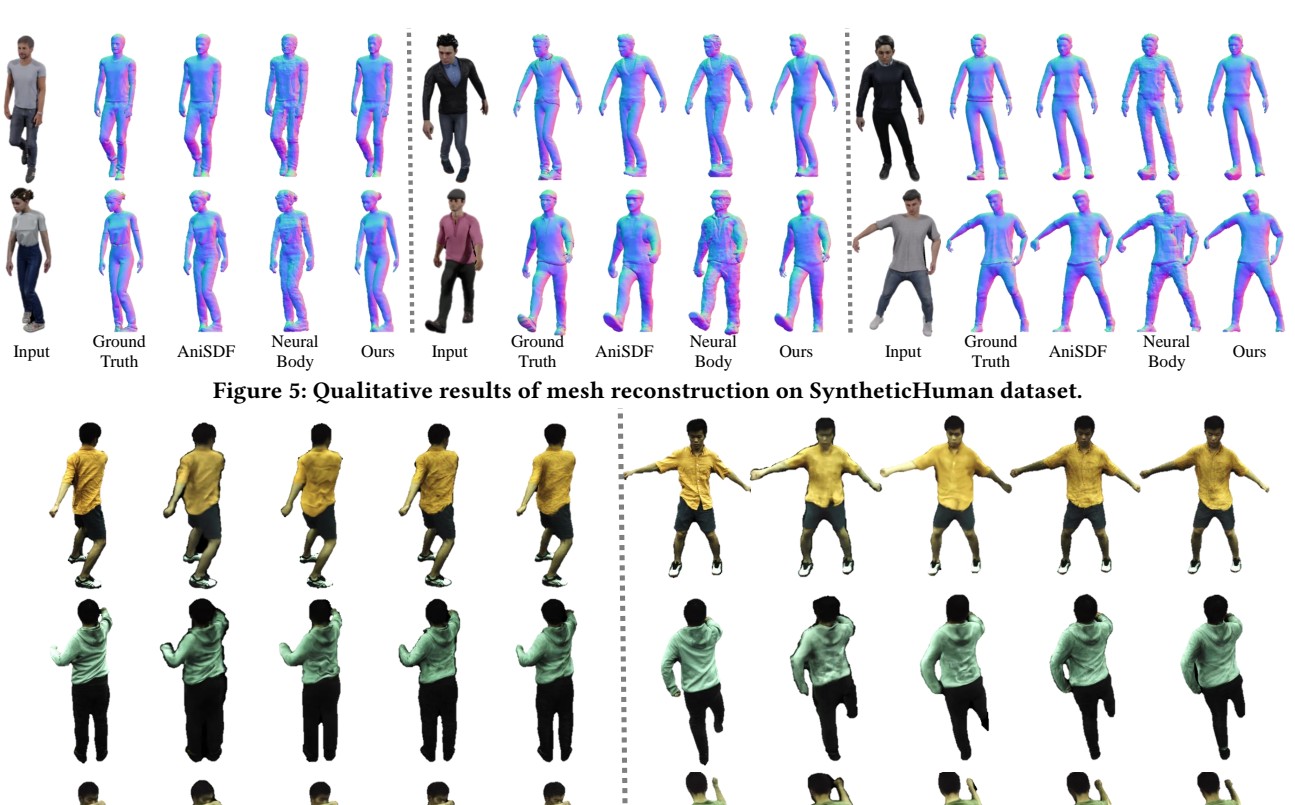

**Figure 5: Qualitative results of mesh reconstruction on SyntheticHuman dataset.**

**Figure 6: Qualitative comparison with SOTA methods in (a) seen poses, and (b) unseen poses on the ZJU-MoCap dataset.**

Qualitative comparison in Fig. 6 showcases the remarkable visual quality of our MCGS (Mesh-Centric Gaussian Splatting) method, particularly in capturing clean and pose-dependent non-linear details. This improvement aligns with the enhancements in LPIPS metric and reinforces the notion that LPIPS is a more representative measure of visual quality compared to PSNR. Neural Body and AniSDF exhibit twisted and oversmoothed renderings, especially for unseen poses. For instance, the renderings in the third row for unseen poses illustrate that Neural Body and AniSDF miss most details on the back. Conversely, the 3DGS-based methods (3DGS-Avatar and ours) generate realistic wrinkles. Our method, in comparison to 3DGS-Avatar, captures more details, as evident in the second row of seen poses, where 3DGS-Avatar produces overly smoothed results while our method represents realistic wrinkles.

## 4.6 Ablation Study

We perform a comprehensive ablation study on subject 387 of ZJU-MoCap dataset, which includes investigating the significance of Mesh-Centric SDF, the importance of learning mesh and appearance dynamics, and the impact of subdividing initialized SMPL mesh. As ZJU-MoCap dataset lacks ground truth mesh, we train the proposed

method using all cameras for a single frame to generate a reference mesh, as depicted in Figure 7. The high accuracy and consistency of the reference mesh with the reference image highlight a promising direction for applying our method to static scene reconstruction.

**i) Mesh-Centric SDF.** We introduce Mesh-Centric SDF in Section 3.3 as a novel surface representation, where the underlying mesh serves as 0 iso-surface and can be optimized using Gaussian Splatting through the generation of Multi-Layer Gaussians from the underlying mesh. This strategy is crucial for capturing precise geometry as shown in Figure 3. To evaluate this, we exclude Mesh-Centric SDF and directly train Mesh-Constrained Gaussians without adjusting opacity $\alpha$ using Equation (14). This variant of the model is denoted as 'w/o Mesh-Centric SDF' in Table 4 and Figure 7, illustrating its inability to learn accurate geometry and only generating a skinned model. This outcome underscores the critical importance of Mesh-Centric SDF in learning precise geometry.

**ii) Learning mesh and appearance dynamics.** We model mesh and appearance dynamics using two MLP networks, denoted as $f_{non-linear}$ and $f_{color}$. By excluding these networks, we introduce entries 'Fixed Mesh' and 'Fixed Appearance' in Table 4 and Figure 7 correspondingly. The comparison in Figure 7 reveals that these two

Table 3: Quantitative results of image rendering on ZJU-MoCap dataset in terms of PSNR (higher is better) and LPIPS (lower is better). NB is short for Neural Body. 3DGS is short for 3DGS-Avatar.

| | Seen Poses | | | | | | | | Unseen Poses | | | | | | | |
|---|---|---|---|---|---|---|---|---|---|---|---|---|---|---|---|---|
| | PSNR ↑ | | | | LPIPS $\times 10^3$ ↓ | | | | PSNR ↑ | | | | LPIPS $\times 10^3$ ↓ | | | |
| | NB | AniSDF | 3DGS | OURS | NB | AniSDF | 3DGS | OURS | NB | AniSDF | 3DGS | OURS | NB | AniSDF | 3DGS | OURS |
| 377 | **33.67** | 32.77 | 32.54 | 32.25 | 31.92 | 32.36 | 18.10 | **17.49** | 30.43 | 31.15 | **31.50** | 31.41 | 38.67 | 33.73 | 19.60 | **18.90** |
| 386 | **36.15** | 34.96 | 35.22 | 34.75 | 31.49 | 36.34 | 24.20 | **21.70** | 32.89 | **33.39** | 33.10 | 32.93 | 44.84 | 39.52 | 27.50 | **25.90** |
| 387 | **31.10** | 30.71 | 30.62 | 30.63 | 48.17 | 49.98 | 30.50 | **28.60** | 28.16 | 28.44 | **28.70** | 28.66 | 55.23 | 51.67 | 32.40 | **31.70** |
| 392 | **35.65** | 34.36 | 34.15 | 33.77 | 39.84 | 43.53 | 24.20 | **23.09** | **31.57** | 30.90 | 31.00 | 31.06 | 51.66 | 48.26 | 30.30 | **29.30** |
| 393 | **33.20** | 31.84 | 31.75 | 31.81 | 45.31 | 45.35 | 27.30 | **25.60** | 28.53 | 28.41 | **29.10** | 28.88 | 58.95 | 52.49 | 34.20 | **33.10** |
| 394 | **34.40** | 33.46 | 33.39 | 33.13 | 41.82 | 40.67 | 23.80 | **22.88** | 29.75 | 29.63 | 30.48 | **30.51** | 55.89 | 47.38 | 29.60 | **28.60** |
| Avg. | **34.03** | 33.02 | 32.95 | 32.72 | 39.76 | 41.37 | 24.68 | **23.23** | 30.22 | 30.32 | **30.65** | 30.58 | 50.87 | 45.51 | 28.93 | **27.92** |

Table 4: PSNR and LPIPS metrics for ablation study.

| | w/o Mesh-Centric SDF | Fixed Mesh | Fixed Appearance | w/o Subdivision | Double Subdivision | Full Model |
|---|---|---|---|---|---|---|
| PSNR | 25.25 | 29.72 | 28.81 | 30.58 | 30.68 | 30.63 |
| LPIPS | 51.27 | 32.82 | 34.20 | 30.03 | 27.98 | 28.60 |

Figure 7: Qualitative results of ablation study.

variants are unable to capture accurate geometry like the hood. Furthermore, Table 4 demonstrates that they exhibit significantly lower PSNR and LPIPS metrics, underscoring the efficacy of our approach in learning mesh and appearance dynamics.

**iii) Subdivision of SMPL mesh.** We subdivide SMPL mesh once to enhance the level of detail in geometry. To evaluate the impact of this strategy, we create two variations termed 'w/o Subdivision' and 'Double Subdivision', where one omits subdivision and the other undergoes subdivision twice. As depicted in Table 4 and Figure 7, increased subdivision results in improved performance both quantitatively and qualitatively. However, excessive subdivision leads to an abundance of faces and Gaussians, as each face is assigned at least one Gaussian. Consequently, the rendering speed drops below real-time (30 FPS). Hence, we opt to perform subdivision only once on SMPL mesh, which already yields superior mesh reconstruction, as evidenced in Table 2, while maintaining real-time efficiency.

## 5 LIMITATIONS AND FUTURE WORK

The Mesh-Centric Gaussian Splatting (MCGS) approach proposed in this work has shown promising results in realistic image rendering and real-time mesh reconstruction. However, there exist certain limitations. Firstly, the reliance on a coarse mesh (such as SMPL in this context) for aligning information across video frames and as an initialization may not be feasible in scenarios involving loose cloth or attachments. One potential avenue for improvement could involve combining a point cloud reconstructed from MVS (similar to the initialization in 3DGS [14]) and constructing an initialized

mesh from the point cloud. Secondly, the optimization process of the underlying mesh does not concern adaptive subdivision or undivision and may encounter challenges when dealing with excessively loose cloth. Exploring the incorporation of adaptive subdivision or undivision algorithm during optimization presents an intriguing direction, which could also extend the proposed method to scene reconstruction and enable direct mesh training from multi-view images. These issues are left for future investigation.

## 6 CONCLUSION

In this paper, we introduce Mesh-Centric Gaussian Splatting (MCGS), a novel method for human avatar modelling with real-time mesh reconstruction from multi-view video. Firstly, we present Mesh-Centric SDF as a unique surface representation, consisting of a thin layer of SDF enveloping an underlying mesh, where the mesh itself serves as 0 iso-surface. This approach eliminates the need for Marching Cubes to extract 0 iso-surface and enables real-time mesh reconstruction. Secondly, we construct Multi-Layer Mesh from the underlying mesh of Mesh-Centric SDF, which acts as sampling layers, and generate Mesh-Constrained Gaussians on these layers. This process facilitates training Mesh-Centric SDF using efficient Gaussian Splatting. Experimental results demonstrate that our method achieves real-time mesh reconstruction and SOTA performance in both image rendering and mesh reconstruction. This research establishes a pathway for direct learning meshes from multi-view videos, offering substantial potential for merging 3D reconstruction research with computer graphics in practical applications.

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
