# OpenReview forum: "Mesh-Centric Gaussian Splatting for Human Avatar Modelling with Real-time Dynamic Mesh Reconstruction"
_acmmm.org/ACMMM/2024/Conference — MM2024 Poster_

### Official Review · Reviewer_ekoC · 2024-05-23

**Rating:** 4
**Confidence:** 3

**Summary:**

This work focuses on human avatar modeling with real-time dynamic mesh reconstruction. It proposes Mesh-Centric Gaussian Splatting (MCGS), which is trained to produce image rendering and mesh from a multi-view video.

MCGS consists of two key parts: Mesh-Centric SDF and Multi-Layer Gaussians. Mesh-Centric SDF can be seen as a counterpart of mesh. Multi-Layer Gaussians are used to optimize Mesh-Centric SDF. Rendering Multi-Layer Gaussians as images, MCGS can optimize Mesh-Centric SDF with image observations. After optimizing, meshes can be extracted from Mesh-Centric SDF at over 30 FPS, which is much faster than existing methods.

**Strengths:**

1. The experiments are sufficient.
2. The mesh reconstruction is quite fast to meet real-time requirements.

**Limitations:**

1. The method section's subtitles are confusing at first glance. I cannot recognize the connections between the method details in the introduction and these subtitles.
2. The details of the method are not clear. I have some questions below:
   1. Why construct multi-layer mesh surfaces from a single SDF layer with random normal vectors? I cannot find the reason to replace a single and accurate mesh surface with multi-layer mesh surfaces.
   2. Are these normal vectors learnable? If not, a set of completely random normals introduces uncertainty to the reconstructed mesh surfaces.
   3. The same questions to the multi-layer design in Gaussian Spaltting.

**Suitability:**

3

---

### Official Review · Reviewer_Fwut · 2024-05-25

**Rating:** 3
**Confidence:** 3

**Summary:**

This study introduces a novel approach, Mesh-Centric Gaussian Splatting (MCGS), which introduces a unique representation Mesh-Centric SDF and optimizes it using high-efficiency Gaussian Splatting。This method can directly optimize the underlying mesh through Gaussian Splatting, providing fast training and rendering speeds derived from Gaussian Splatting, as well as precise surface learning of SDF.

**Strengths:**

This study presents an innovative approach called Mesh-Centric Gaussian Splatting (MCGS), which introduces a novel Mesh-Centric Signed Distance Function (SDF) representation. This new representation envelops the underlying mesh with a thin layer of SDF, which can be efficiently derived from the mesh itself. This method enables mesh optimization via SDF, using the mesh as the zero iso-surface and eliminating the need for the slower Marching Cubes algorithm.
The primary contribution of this study is the introduction of the Mesh-Centric SDF. By deriving SDF from the mesh, the process allows for direct optimization of the mesh through SDF representation. The secondary contribution is the optimization of this Mesh-Centric SDF using high-efficiency Gaussian Splatting.
In detail, the approach involves dispersing the underlying mesh of the Mesh-Centric SDF into multiple layers, then generating Mesh-Constrained Gaussians within these layers to create Multi-Layer Gaussians. These Mesh-Constrained Gaussians are confined within a 2D surface space defined by the mesh, ensuring precise alignment between Gaussian rendering and the mesh geometry.

**Limitations:**

1. The article only uses two datasets. It is recommended to add another dataset for validation and analysis of the model. Besides analyzing the advantages of your model, please also discuss its disadvantages.
2. It is suggested to describe the technical details of your implementation more thoroughly, so interested researchers can replicate your work.
3. The Mesh-Centric Gaussian Splatting (MCGS) method presented in the article shows significant advantages in real-time mesh reconstruction and image rendering. However, the comparative experiments should more thoroughly discuss the comparisons with other methods, including specific performance metrics and experimental results.
4. The description of the Mesh-Centric SDF and Gaussian Splatting optimization process is unclear and lacks sufficient detail and explanation. This may prevent readers from accurately understanding the implementation steps of the method.
5. When introducing the new method, the article does not sufficiently discuss its distinctions and innovations compared to existing research. The lack of comparative analysis may make it difficult for readers to understand the advantages of this method over existing techniques.

**Suitability:**

2

---

### Official Review · Reviewer_Gvua · 2024-05-26

**Rating:** 4
**Confidence:** 4

**Summary:**

The paper introduces MCGS for real-time mesh reconstruction of human avatars from multi-view video. This method uses a Mesh-Centric Signed Distance Field (SDF) representation, which envelops the underlying mesh, and optimizes it through Gaussian Splatting. By dispersing the underlying mesh into multiple layers and generating Mesh-Constrained Gaussians, MCGS effectively aligns Gaussian rendering with mesh geometry, allowing for precise surface learning. Experimental results demonstrate that MCGS achieves over 30 FPS in dynamic mesh reconstruction.

**Strengths:**

- The proposed method achieves dynamic mesh reconstruction at over 30 FPS and outperforms traditional SDF methods that rely on Marching Cubes.

- The introduced Mesh-Centric Signed Distance Field (SDF) is a novel surface representation

- The proposed method has notable better quality than baselines

**Limitations:**

- It would be better if the authors can show some video results.
- From Table 4., It seems that without subdivision, the results are better than the final model, why the authors are still using that?

**Suitability:**

3

---

### Official Review · Reviewer_F9FJ · 2024-06-06

**Rating:** 4
**Confidence:** 3

**Summary:**

The paper introduces a novel technique to optimize human body meshes obtained using off-the-shelf pose estimators and semantic segmentation models from multiple views. A pose-dependent deformation model deforms the template mesh in canonical pose, and then the deformed template is skinned in the estimated pose using LBS. The paper introduces multi-layer meshes constructed by deforming vertices along their normals. Gaussians are attached to these multi-layer meshes, making it renderable. The deformation model and the gaussians are optimized to reduce photo-consistency and mask-based losses with regularization.

**Strengths:**

The paper is well written and easy to follow.
Conceptually if an SDF can be derived using a skeleton or a coarse mesh then the method can be used to get high quality output.
The approach seems to produce high quality human meshes compared to other approaches.
LPIP values for the generated mesh renders are better than comparable methods.

**Limitations:**

The work relies on SMPL or some kind of a prior mesh for any of it to work. I feel this should be clearly shown in the Fig 1, so as to not over emphasizing the methods capabilities.

Since the approach needs a coarse mesh to begin with the usefulness of this approach compared to 3DGS+MC or 3DGS combined with a meshing technique that doesn't require a reference/coarse initial mesh is greatly reduced.

Given that the Gaussians here represent flat surfaces and the fact that the surface being optimized is an explicitly defined mesh, why did the authors pick gaussians as the preferred primitive? Would it have been more straightforward to use differentiable rasterization in this case?

Comparisons on novel view synthesis with basic 3DGS, say, are missing. It would be informative to see if optimizing for a mesh also improves output image quality.

The paper claims the deformation model as one of the contributions, but it does not seem to make too big of a difference. Related, SMPL includes its own pose-dependent shape correction term (pose blend-shapes) to deal with skinning artifacts. Did the authors include this term in the ablation without the deformation model ("Fixed mesh" if I understand correctly)?

Typos:
Table 1 and Figure 1 -- Incorrectly bolded numbers
line 316 -- Prelininary -> Preliminary

**Suitability:**

3

---

### Meta-Review · Area_Chair_B2pp · 2024-06-29

**Recommendation:** Accept (Poster)
**Confidence:** 5

**Metareview:**

This paper focuses on human avatar modeling with real-time dynamic mesh reconstruction via the proposed mesh-centric Gaussian Splatting. It can extract mesh from Mesh-Centric SDF at over 30FPS, which is much faster than existing methods.

This paper accept three final borderline accepts and one initial Borderline reject. The reviewer who rates as Borderline reject didn't update the final recommendation. I read the author's rebuttal, and found that the rebuttal well address the third reviewer's concern. Therefore the paper can be accepted for publication at ACM MM.